# Suicidal Ideation among University Students: A Moderated Mediation Model Considering Attachment, Personality, and Sex

**DOI:** 10.3390/ijerph19106167

**Published:** 2022-05-19

**Authors:** Antonella Granieri, Silvia Casale, Maria Domenica Sauta, Isabella Giulia Franzoi

**Affiliations:** 1Department of Psychology, University of Turin, via Verdi 10, 10124 Turin, Italy; antonella.granieri@unito.it (A.G.); mariadomenica.sauta@unito.it (M.D.S.); 2Department of Health Sciences, University of Florence, Viale Pieraccini 6, 50139 Florence, Italy; silvia.casale@unifi.it

**Keywords:** attachment, negative affectivity, personality, university students, suicidal ideation

## Abstract

The present study aimed to examine the link between attachment, personality traits, and suicidal ideation with consideration of the potential moderating role of age and sex. The Suicidal History Self-Rating Screening Scale, the Personality Inventory for DSM-5-Brief Form, and the Attachment Style Questionnaire were administered to 183 students. There was a significant indirect effect of need for approval on suicidal ideation via detachment. Moreover, the moderated mediation models of need for approval and preoccupation with relationships on suicidality via negative affectivity were significant in men, but not women, whereas the moderated mediation model of need for approval on suicidality via detachment was significant in women, but not men. Young men and women seem at risk for increased suicidal ideation based on specific attachment and personality characteristics, which should be considered for the development of prevention and therapeutic interventions.

## 1. Introduction

According to the World Health Organization [1,2], suicide deaths in 2019 amounted to over 700,000 worldwide, with rates in men being almost double those in women. Notably, despite having the most dramatic decline of over 40% during the past two decades, suicide rates in the European region were the highest among all of the WHO regions, which were 21.9 per 100,000 population in 2000 and 12.8 per 100,000 population in 2019. In Italy, the crude death rate for suicide in 2019 was 6.7 per 100,000 population. In particular, the statistics for causes of death gathered by the Italian National Institute of Statistics [3] in the last available year detected 3820 deaths because of suicide in 2018, of which 253 (6.6%) were in young adults aged 20–29 years. Moreover, men comprised 79% of the young adults dying due to suicide, which was the third most common cause of death, after accidents and malignant tumors [3].

Previous research has focused on suicidal behavior among young adults [4], and, in particular, university students [5,6], who are known to experience psychological distress and mental health problems at higher rates than do people in the general population and community peers [7,8,9]. Indeed, previous research has reported that 7.6% to 15% of university students have experienced suicidal behavior at least once in the past [10,11,12], and 15.1% to 16.2% of students have had suicidal thoughts at least once in their lifetime [11,13]. Moreover, current suicidal ideation is experienced by 20% of students [10], and suicidal risk by 13.1% [14].

Attachment theory offers a possible model to explain suicidal ideation and suicidal attempts based on early adverse parenting experiences resulting in the development of insecure attachments, which could act as a vulnerability factor for suicide risk later in adolescence or young adulthood [15]. Empirical evidence supports this perspective by showing that secure attachment protects against suicidal ideation [16]. In contrast, individuals who exhibit insecure attachment styles are at an increased risk of suicidality [17,18,19,20,21,22]. Specifically, the link between anxious attachment and suicidal ideation has been extensively investigated with results consistently showing a positive association in clinical samples [19]. A positive link has also been confirmed in undergraduate and graduate students [23,24,25]. A recent review by Zortea and colleagues [26] revealed higher levels of attachment anxiety in groups that experience suicidal ideation compared with controls, and a positive correlation between the level of attachment anxiety and suicidal ideation. Specifically, Riggs and Jacobvitz [27] showed that individuals in the anxious group reported the highest suicidal ideation.

To date, the potential mediators and moderators of the relationship between anxious/preoccupied attachment and suicidal ideation have been rarely investigated. In terms of moderators, the review by Zortea and colleagues [26] highlighted that very few studies investigating the relationship between suicidal ideation and insecure attachments controlled for age and sex, and claimed that further research is needed on this topic. Zeyrek and colleagues [25] examined the perceived likelihood of future suicidal behavior among university students and found that it was significantly correlated with preoccupied attachment style in women. Moreover, sex differences in suicidal intention and behavior have been reported worldwide [28,29,30,31].

Regarding mediators, attachment style has been shown to influence emotion regulation and the processing of negative affectivity [32,33]. Specifically, there is a positive relationship between an anxious attachment style and negative affectivity [34], and research has demonstrated that individuals with negative affectivity are at increased risk of suicidal ideation [35,36]. Moreover, suicidal ideation can be related to alexithymia and its interplay with emotion regulation. Indeed, alexithymia can leave young adults more vulnerable and cause psychological distress with subsequent suicidal ideation [37,38]. In such contexts, it is important to recognize alexithymic traits early and to design psychological treatments specifically focused on affective regulation [39,40].

Attachment styles were also shown to influence detachment and disconnection from interpersonal relationships [41]. Research showed a positive relationship between an anxious attachment style and withdrawal [42,43,44], and individuals with higher levels of social disconnection and loneliness are at increased risk of suicidal ideation [45,46,47]. Taken together, these findings suggest that negative affectivity and detachment mediate the link between insecure attachment and suicidal ideation.

In addition, recent research suggested that suicidal ideation and suicidal attempts have distinct risk and protective factors. For example, women show higher suicidality than men, but are less likely to commit suicide [2,48,49].

### Present Study

In the current study, we sought to extend the literature on suicidal ideation in various respects. First, we re-examined the link between the anxious/preoccupied attachment dimensions (i.e., the need for approval and preoccupation with relationships) and suicidal ideation, with consideration of the potential moderating roles of age and sex. Second, we aimed to build upon previous results on the association between anxious/preoccupied attachment and suicidal ideation by exploring the mediating roles of negative affectivity and detachment. To the best of our knowledge, no study has investigated the relationship between the anxious/preoccupied attachment dimensions, negative affectivity, detachment, and suicidal ideation in university students by also controlling for sociodemographic characteristics.

Thus, we tested the following hypotheses:

**Hypothesis** **1** **(H1).**
*University students’ suicidal ideation is correlated with sociodemographic characteristics (i.e., age and sex), the need for approval, preoccupation with relationships, negative affectivity, and detachment;*


**Hypothesis** **2** **(H2).**
*Personality dimensions (i.e., negative affectivity and detachment) mediate the relationship between attachment dimensions (i.e., need for approval and preoccupation with relationships) and suicidal ideation;*


**Hypothesis** **3** **(H3).**
*Sex moderates the association between negative affectivity and detachment, and suicidal ideation.*


## 2. Materials and Methods

### 2.1. Study Design and Participants

This study was a descriptive cross-sectional study. Undergraduate students attending the University of Turin (UniTo) were recruited between October 2018 and February 2020, and enrollment was ended prematurely because of the spread of the COVID-19 pandemic. Inclusion criteria were participants aged between 18 and 29 years (emerging adults). The exclusion criterion was having a poor knowledge of the Italian language.

Enrollment was conducted through cooperation with professors, student representatives, departmental councils, and Heads of Departments. We contacted students in 35 (50.73%) out of the 69 bachelor’s degree courses and four (44.44%) out of the nine single-cycle master’s degree courses available at UniTo. Nine hundred and seventy-nine students indicated a willingness to participate in the study. Of these, only 183 (18.69%; women = 71.58%; mean age = 21.54 ± 2.15 years) students agreed to be tested. The remaining students were no longer available when contacted to arrange the study appointment. State and trait anxiety data of a subgroup of students were published previously [50]. 

### 2.2. Measures

The study involved the administration in pencil and paper of a survey, which included a series of self-report measures validated for the Italian population. The present research focuses on scores obtained from the Suicidal History Self-Rating Screening Scale (SHSS) [51], the Personality Inventory for DSM-5-Brief Form (PID-5-BF) [52,53], and the Attachment Style Questionnaire (ASQ) [54,55].

The ASQ is a 40-item self-report measure aimed at assessing adult attachment. It comprises five scales: Confidence (C); Relationship as Secondary (RS); Discomfort with Closeness (DC); Need for Approval (NA); and Preoccupation with Relationship (PR). Each item is rated on a six-point scale (ranging from 1 = ‘totally disagree’ to 6 = ‘totally agree’). For our study, we only analyzed the NA and the PR scales. The cut-off scores for the Italian sample were >25 for NA, which indicates an excessive need to gain approval, support, and responsiveness from others, and >33 for PR, which indicates an excessive strive for personal contact [56]. In our study, Cronbach’s alphas were 0.77 for NA and 0.71 for PR. 

The SHSS is a 16-item measure assessing the propensity for suicide in terms of thoughts of death, suicidal ideation, and behavior. Participants were asked to answer eight yes/no questions based on the previous 12 months and eight yes/no questions based on their lifetime excluding the previous 12 months. For our study, we only used the total score. Higher total scores indicate more severe suicidal ideation, and scores > 8 indicate a high risk of suicidal behavior. In our study, Cronbach’s alpha was 0.88 for the total score.

The PID-5-BF is a 25-item dimensional self-report measure assessing five broad pathological personality traits: Negative Affectivity (NegA); Detachment (DE); Antagonism (A); Disinhibition (DI); and Psychoticism (P). Items were derived from the 220 items of the self-report Personality Inventory for DSM-5 [52,53]. Participants were asked to rate how accurately each item described themselves on a four-point scale (ranging from 0 = ‘very false or often false’ to 3 = ‘very true or often true’). Mean scores reflected the presence of each trait and overall personality dysfunction on a scale from 0 = ‘absent’ to 3 = ‘severe’. For our study, we only included the NegA and DE scales. Cronbach’s alphas were 0.66 for NegA and 0.71 for DE.

The scales were administered at the university in the presence of a psychologist or a trained post-graduate psychology student. The average time of completion was 39.79 ± 9.89 min (range 20–77 min).

### 2.3. Statistical Analysis

Data analyses were conducted using the Statistical Package for the Social Sciences (SPSS; IBM Corp., Armonk, NY, USA), version 27. First, descriptive statistics of the study variables were calculated. We then performed point-biserial and Pearson correlations to obtain an initial overview of the variables to be included in our moderated mediation model. All tests were two-tailed, and statistical significance was set at *p* ≤ 0.05. Finally, we conducted a moderated mediation analysis using the PROCESS macro for SPSS (version 3.5) [57] using model 15. We set suicidal ideation as a dependent variable, NA and PR as independent variables, NegA and DE as mediating variables, sex as a moderator, and age as a covariate. The PROCESS dialog box is set for only one independent variable; however, as suggested by Hayes [57], the direct and indirect effects of more than one X variable can be estimated by executing PROCESS k times, each time entering one independent variable in the model as X and the remaining independent variables as covariates. All resulting paths and direct and indirect effects will equate to all being estimated simultaneously. The direct and indirect effects were estimated using Preacher and Hayes’ [58] bias-corrected non-parametric bootstrapping techniques with 5000 bootstrap samples. We used the mean center for the construction of products. As suggested in previous studies [59], the significance of the mediation and moderated mediation effects were evaluated using 95% bias-corrected confidence intervals (CIs). If the CIs did not contain a zero, the effects were considered statistically significant.

## 3. Results

### 3.1. Sociodemographic and Clinical Characteristics of the Sample

The sample consists of 184 university students (71.58% females), aged between 18 and 29 (M = 21.54; SD = 0.15). Students’ mean scores suggested that the levels of NA (M = 23.12; SD = 6.43) and PR (M = 29.89; SD = 6.30) were in line with the normative data for the Italian population. However, 31.15% and 28.42% of the participants scored higher than the Italian cut-off scores for what concerns NA and PR, respectively. Moreover, students’ mean scores reflected mild levels of NegA (M = 1.39; SD = 0.62) and DE (M = 0.74; SD = 0.57). Regarding suicidal ideation, the mean SHSS score indicated not-at-risk (M = 1.98; SD = 2.86), and 3.28% of the sample was at risk of suicidal behaviors. 

### 3.2. Preliminary Analyses for the Moderated Mediation Model

As expected (Table 1), suicidal ideation correlated moderately and positively with NA, PR, NegA, and DE. In addition, both NA and PR showed a moderate positive correlation with NegA and DE. Weak correlations were found between sex and NegA and DE. Weak negative correlations were also found among PR, NegA, and DE. Moreover, there was a weak negative correlation between age and sex.

### 3.3. Moderated Mediation Analysis

As reported in Table 2, the regression analysis for NegA, controlling for age, showed a significant positive effect of NA and PR. The interaction between sex and NA was not statistically significant. Overall, the predictors explained 33% of the variance observed in the NegA scores (F_(3179)_ = 29.95, *p* < 0.001).

Controlling for age, we found a significant positive effect of NA on DE, but no significant effect of PR. Overall, the predictors explained 14% of the variance observed in the DE scores.

As shown in Table 3, controlling for age did not reveal any significant effects of NA or PR on suicidal ideation. 

However, we found a positive and significant effect of NegA and DE on suicidal ideation. Sex was not a significant predictor of suicidal ideation. Moreover, we did not find significant effects of the interaction between sex and NA or sex and PR on suicidal ideation. However, the effects of the interaction between sex and NegA and sex and DE on suicidal ideation were significant. Overall, the predictors explained 28% of the variance observed in suicidal ideation (F_(9173)_ = 7.345, *p* < 0.001). The inclusion of the interaction between sex and NegA in the regression model led to a change in R^2^ = 0.017 (F_(1173)_ = 4.117, *p* = 0.044), and the inclusion of the interaction between sex and DE led to a change in R^2^ = 0.026 (F_(1173)_ = 6.178, *p* = 0.014).

As shown in Table 4, we did not find a direct effect of NA and PR on suicidal ideation in men or women.

For the moderators, the simple slope analysis (Figure 1) of the interaction model showed a significant positive relationship between NegA and suicidal ideation in men (β = 2.186, standard error (SE) = 0.636, *p* =.001) but not in women (β = 0.473, SE = 0.438, *p* = 0.282).

Moreover, we found a significant positive relationship between DE and suicidal ideation in women (β = 1.804, SE = 0.441, *p* < 0.001) but not in men (β = 0.193, SE = 0.616, *p* = 0.755; Figure 2).

As shown in Table 5, the moderated mediation model of NA on suicidal ideation through NegA was significant in men, but not in women. Overall, the moderated mediation model was not significant. Moreover, the moderated mediation model of NA on suicidal ideation through DE was significant in women, but not in men. Overall, this moderated mediation model was significant.

As shown in Table 6, the moderated mediation model of PR on suicidal ideation through NegA was significant in men, but not in women, whereas the moderated mediation model of PR on suicidal ideation through DE was not significant in men or women. Overall, neither of the two moderated mediation models was significant.

Figure 3 illustrates the final moderated mediation model with its coefficients.

## 4. Discussion

This study examined suicidal ideation in undergraduate students by examining its association with sex, anxious/preoccupied attachment dimensions (i.e., NA and PR), and personality dimensions (i.e., NegA and DE).

Our data suggested a lower suicidal risk in our sample than that found in previous studies on university students, although this discrepancy may be due to differences in outcome measures [10,11,13,14]. Nevertheless, we cannot neglect the risk of suicidal behaviors observed in 3.28% of our sample.

Concerning our first hypothesis, consistent with the literature, suicidal ideation in students was positively correlated with NA, PR, NegA, and DE [35,36,60,61]. Moreover, both NA and PR showed a positive and significant correlation with NegA and DE. 

In contrast to previous research, sex was only negatively correlated with NegA and DE, but not with personality dimensions [62,63,64]. For the hypothesized covariate, age correlated negatively with PR, negative affect, and DE, as expected [65,66,67].

With respect to our moderated mediation models, we found a significant positive effect of NA and PR on NegA, and a significant positive effect of NA on DE, which was in line with our second hypothesis. This is consistent with previous research showing that aversive mood states and difficulties in emotion regulation are strongly related to suicidal ideation [68]. Moreover, previous research has highlighted that individuals who grew up in difficult caregiving environments with corresponding insecure attachment representations often show deficits in the development of affect regulation strategies [69,70,71], and are more likely to exhibit a history of suicidal ideation [72]. 

For our moderated mediation models, we also found a significant positive effect of NegA and DE on suicidal ideation, which suggested that frequent and intense experiences of negative emotions (i.e., guilt, fear, or shame), avoidance of social-emotional experiences, and withdrawal from interpersonal relationships lead to impaired vitality and a lowered investment in life. 

Contrary to previous research [26], we found no significant direct effects of attachment dimensions (NA and PR) on suicidal ideation, and the only overall significant indirect effect was that of NA on suicidal ideation via DE. Furthermore, surprisingly and in contrast to previous studies [25,73,74], sex was not a significant predictor of suicidal ideation, and we did not reveal significant effects of the interaction between sex and NA, and sex and PR.

However, the role of sex was crucial when we considered the effect of its interaction with personality dimensions (NegA and DE) on suicidal ideation. Indeed, we found a significant positive association between NegA and suicidality in men, but not in women, and a significant positive association between DE and suicidality in women, but not in men. Moreover, the moderated mediation models of NA and PR on suicidality via NegA were significant in men, but not in women, whereas the moderated mediation model of NA on suicidality via DE was significant in women, but not in men.

Such results confirm that suicidal ideation in young adults is complex and multidimensional [75]. During the emerging adulthood stage, young women and men must develop a stable adult identity, morals and ethics, harmonious family and group relationships, engagement in the community, educational attainment, and capabilities to confront and act on reality. This process can be profoundly undermined and derailed by unresolved conflicts that are linked to previous developmental stages and earlier disharmonies, and also by relational, economical, academic, and occupational difficulties [76]. Affective flooding and regression toward more primitive modes of thought, defenses, and modes of relating are common [77]. Moreover, emerging adults can experience important fluctuations in recognizing their genuine qualities and capacities, along with the feeling of consistency between who they feel they are and how they are experienced by others [77,78]. Their relative lack of life experiences, combined with idealized aspirational fantasies, and the feeling that every decision changes the course of their future, can lead to a sense of personal crisis, severe judgment, and attack of the self, which heightens their vulnerability to suicide [77,79].

In this context, our study suggests that different paths lead to higher suicidal risk depending on the sex of the emerging adult. For men, suicidality was increased by NegA, which also mediated the impact of NA and PR on suicidality. The need to gain approval, support, and responsiveness from others, as well as the strive for personal contact may improve suicidal risk indirectly, but they may also increase the tendency to experience unpleasant feelings, lability, or restricted emotionality. Therefore, suicidal ideation can be considered an extreme and dysregulated attempt to deal with these feelings and seek help [75].

In contrast, for women, suicidal ideation was increased by DE, which also mediated the impact of NA on suicidality. Thus, negative self-image and the strive for personal contact may heighten suicidal risk indirectly, but may also increase a women’s need to protect themselves from the risk of being engulfed or abandoned in their intimate relationships, which leads to interpersonal withdrawal and isolation. This feeling of disconnection from significant others and the external world can lead to a sense of mental disintegration, which sustains suicidal ideation [80].

Further investigation on the relationship between attachment styles, suicidal ideation, and affective regulation may provide a more comprehensive picture of the plight of university students. Moreover, it may help to develop and implement effective programs and training strategies to address suicidal ideation by considering specific risk factors connected to sex [81,82,83].

### Limitations and Future Directions

This study has several critical limitations. First, we enrolled a convenience sample of Italian undergraduate students from only one university, which limits the generalizability of the present results. Future studies should involve community and clinical samples. Second, some of the scales included in our model, particularly the NegA scale, had limited reliability. Furthermore, the cross-sectional design did not allow for causal inferences to be made; theoretically, it is plausible that attachment styles and personality predict suicidal ideation, rather than the opposite. In any case, we should be cautious in interpreting the present findings as supporting the existence of predictive links between the studied variables and suicidal ideation. Further longitudinal studies are needed to explore the development of university students’ distress over time and its association with other clinical and social variables. Moreover, psychological variables were assessed using self-report measures; as such, future studies should also consider clinical and observational data. Finally, in our study we considered the sex and not the gender variable. Indeed, few studies on suicidal ideation investigated gender-related facets from social, psychological, and cultural perspectives [84,85]. Thus, future research should also consider gender differences and consider their impact on suicide risk.

## 5. Conclusions

Despite these limitations, the present study is the first to explore suicidal ideation in undergraduate students, focusing on anxious/preoccupied attachment (i.e., NA and PR) and personality dimensions (i.e., NegA and DE) and considering sociodemographic characteristics. There is a growing consensus that more comprehensive services should be provided to support students with mental health concerns [86]: our findings strongly suggest that young men and women are at an increased risk of suicidal ideation depending on different attachment and personality characteristics, which must be considered for the development of prevention and therapeutic interventions targeted at university students. In particular, difficulties in managing negative emotions were shown to be a critical factor for young men, whereas young women who experience interpersonal withdrawal and mistrust were particularly vulnerable to suicide. Thus, although it is not only emerging adults with these characteristics that must be considered at higher risk of suicidal ideation, interventions should target emotion regulation strategies, particularly in young men, and recovering the possibility of an investment in the external world, particularly in young women.

## Figures and Tables

**Figure 1 ijerph-19-06167-f001:**
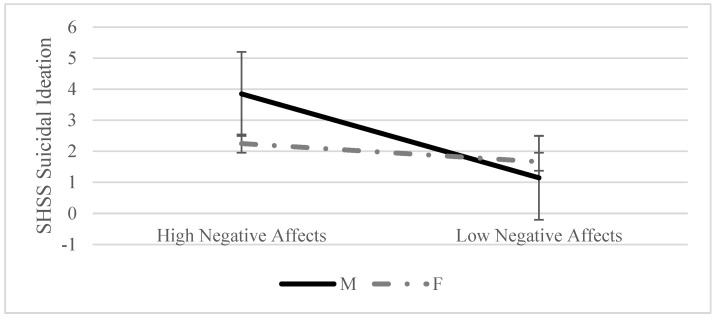
Simple slope analysis for negative affectivity.

**Figure 2 ijerph-19-06167-f002:**
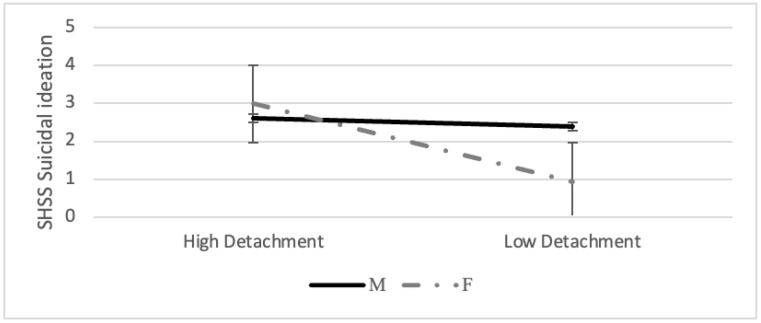
Simple slope analysis for detachment.

**Figure 3 ijerph-19-06167-f003:**
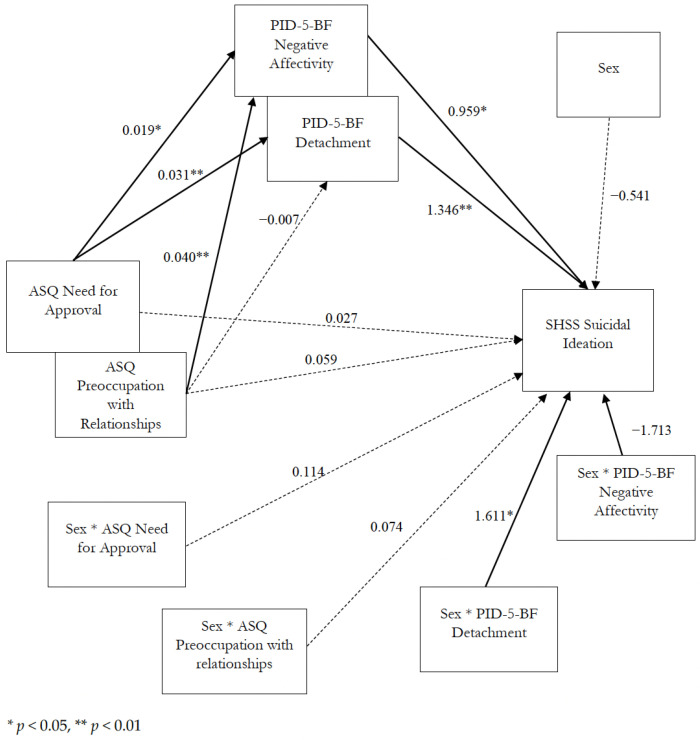
Coefficients of the moderated mediation model. Dotted lines indicate no statistically significant predictors, and continuous lines indicate statistically significant predictors.

**Table 1 ijerph-19-06167-t001:** Pearson and point-biserial correlations.

	SHSS Suicidal Ideation	ASQ Need for Approval	ASQ Preoccupation with Relationships	PID-5-BF Negative Affectivity	PID-5-BF Detachment	Age
Pearson correlations						
ASQ Need for Approval	0.322 **					
ASQ Preoccupation with relationships	0.317 **	0.553 **				
PID-5-BF Negative Affectivity	0.378 **	0.441 **	0.539 **			
PID-5-BF Detachment	0.358 **	0.330 **	0.148 *	0.247 **		
Age	−0.142	−0.123	−0.179 *	−0.219 **	−0.208 **	
Point-biserial correlations						
Sex	−0.016	0.074	0.131	0.190 *	−0.149 *	−0.147 *

* *p* < 0.05, ** *p* < 0.01.

**Table 2 ijerph-19-06167-t002:** Moderated mediation model analysis for negative affectivity and detachment.

Outcome Variables	Independent Variables	β	S.E.	t	*p*	95% CI
**PID-5-BF Negative Affectivity**						
	ASQ Need for Approval	0.019	0.007	2.752	0.007	[0.005; 0.033]
	ASQ Preoccupation with Relationships	0.040	0.007	5.493	<0.001	[0.025; 0.054]
	Age	−0.035	0.018	−1.972	0.050	[−0.071; 0.000]
**PID-5-BF Detachment**						
	ASQ Need for Approval	0.031	0.007	4.227	<0.001	[0.017; 0.046]
	ASQ Preoccupation with Relationships	−0.007	0.008	−0.934	0.351	[−0.022; 0.008]
	Age	−0.047	0.019	−2.532	0.012	[−0.084; −0.010]

**Table 3 ijerph-19-06167-t003:** Moderated mediation model analysis for suicidal ideation.

Outcome Variables	Independent Variables	β	S.E.	t	*p*	95% CI
**SHSS Suicidal Ideation**						
	ASQ Need for Approval	0.027	0.037	0.738	0.462	[−045; 0.099]
	ASQ Preoccupation with Relationships	0.059	0.039	1.520	0.130	[−0.018; 0.136]
	PID-5-BF Negative Affectivity	0.959	0.375	2.559	0.011	[0.220; 1.699]
	PID-5-BF Detachment	1.346	0.363	3.706	<0.001	[0.629; 2.063]
	Sex	−0.541	0.441	−1.226	0.222	[−1.411; 0.329]
	Age	−0.042	0.091	−0.465	0.643	[−0.223; 0.138]
	Sex * ASQ Need for Approval	0.114	0.074	1.544	0.125	[−0.032; 0.360]
	Sex * ASQ Preoccupation with relationships	0.074	0.082	0.905	0.367	[−0.087; 0.235]
	Sex * PID-5-BF Negative Affectivity	−1.713	0.740	−2.314	0.022	[−3.174; −0.252]
	Sex * PID-5-BF Detachment	1.611	0.752	2.144	0.033	[0.128; 3.094]

“*” indicates the interaction between the two variables.

**Table 4 ijerph-19-06167-t004:** Conditional direct effects of need for approval and preoccupation with relationships on suicidal ideation.

	β	S.E.	t	*p*	95% CI
**Conditional direct effect of need for approval on suicidal ideation**					
Men	−0.055	0.065	−0.845	0.399	[−0.183; 0.073]
Women	0.060	0.042	1.416	0.159	[−0.023; 0.142]
**Conditional direct effect of preoccupation with relationships on suicidal ideation**					
Men	0.006	0.073	0.088	0.930	[−0.138; 0.151]
Women	0.080	0.044	1.843	0.067	[−0.006; 0.166]

**Table 5 ijerph-19-06167-t005:** Indirect effects of need for approval on suicidal ideation.

	β	Bootstrap S.E.	Bootstrap 95% CI
**Conditional indirect effects of need for approval on suicidal ideation via negative affectivity for men vs. women**			
Men	0.042	0.025	[0.003; 0.098]
Women	0.009	0.011	[−0.007; 0.036]
**Index of moderated mediation**			
	−0.033	0.023	[−0.082; 0.005]
**Conditional indirect effects of need for approval on suicidal ideation via detachment for men vs. women**			
Men	0.006	0.020	[−0.030; 0.052]
Women	0.056	0.021	[0.022; 0.103]
**Index of moderated mediation**			
	−0.050	0.027	[0.003; 0.111]

**Table 6 ijerph-19-06167-t006:** Indirect effects of preoccupation with relationships on suicidal ideation.

	β	Bootstrap S.E.	Bootstrap 95% CI
**Conditional indirect effects of preoccupation with relationships on suicidal ideation via negative affectivity for men vs. women**			
Men	0.085	0.042	[0.008; 0.170]
Women	0.020	0.019	[−0.017; 0.058]
**Index of moderated mediation**			
	−0.066	0.045	[−0.152; 0.021]
**Conditional indirect effects of preoccupation with relationships on suicidal ideation via detachment for men vs. women**			
Men	−0.001	0.007	[−0.016; 0.013]
Women	−0.013	0.018	[−0.049; 0.020]
**Index of moderated mediation**			
	−0.013	0.018	[−0.053; 0.020]

## Data Availability

The data that support the findings of this study are available on request from the corresponding author. The data are not publicly available due to privacy or ethical restrictions.

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
