# Peer review of "Suicidal Ideation among University Students: A Moderated Mediation Model Considering Attachment, Personality, and Sex"

_ijerph, 2022, doi:10.3390/ijerph19106167_

Round 1

Reviewer 1 Report

I believe that the issue of suicide in the young population is a current issue with great social relevance.

The authors provide concise information on the subject.

I have some comments and/or qüestions about this research:

I have a question about the administration of the questionnaires: were they administered in pencil and paper or online?

Was the research approved by the university ethics committee? No mention is made.

I would appreciate an explicit description of it. The study addresses the gender variable, but from a social, psychosocial, and cultural point of view, there is a wider range of possibilities for using the gender variable, and it would be interesting to add it in the future because it covers more psychological aspects than the sex variable. I understand that the authors have followed the same line as previous research on the subject of suicide.

Finally, adapt the references to the format of the journal.

Reviewer 2 Report

In the current study, we sought to extend the literature on suicidal ideation in various aspects. First, the Author re-examined the link between anxious/preoccupied attachment dimensions (i.e., the need for approval and preoccupation with relationships) and suicide ideation with consideration of the potential moderating roles of age and sex. Second, They aimed to build upon previous results on the association between anxious/preoccupied attachment and suicide ideation by exploring the mediating roles of negative affectivity and detachment. 

Overall, I found the present study timely, innovatie, well conducted and scientifically sound. However, I have some suggestions aimed to improve the quality of the paper and these are outlined below:

1) In the introduction, it should be noted that suicide ideation might be orginated in such saqmple of students from a combination of alexythimia and emotional dysregulation. Please, discuss this point with appropriate references (see dois: 10.3390/brainsci10090591).

2) Please note that this was a convenience sample and report it in the limitations.

3) I believe that table 1 can be condensed in the text.

4) In my opinion Pearson's correlations are generally quite useless. I prefer partial correlations controlling for several potential confounding variables.
